# Impact of ACE and Endoplasmic Reticulum Aminopeptidases Polymorphisms on COVID-19 Outcome

**DOI:** 10.3390/diagnostics13020305

**Published:** 2023-01-13

**Authors:** Amany A. Ghazy, Abdulrahman H. Almaeen, Ibrahim A. Taher, Abdullah N. Alrasheedi, Amel Elsheredy

**Affiliations:** 1Department of Pathology, Microbiology and Immunology Division, College of Medicine, Jouf University, Sakaka 72388, Saudi Arabia; 2Department of Pathology, College of Medicine, Jouf University, Sakaka 72388, Saudi Arabia; 3Department of Otolaryngology/Head & Neck Surgery, College of Medicine, Jouf University, Sakaka 72388, Saudi Arabia; 4Department Microbiology, Medical Research Institute, Alexandria University, Alexandria 5422004, Egypt

**Keywords:** ACErs4291, ERAP1rs26618, COVID-19 outcome

## Abstract

Background: COVID-19 outcomes display multiple unexpected varieties, ranging from unnoticed symptomless infection to death, without any previous alarm or known aggravating factors. Aim: To appraise the impact of ACErs4291(A/T) and ERAP1rs26618(T/C) human polymorphisms on the outcome of COVID-19. Subjects and methods: In total, 240 individuals were enrolled in the study (80 with severe manifestations, 80 with mild manifestations, and 80 healthy persons). ACErs4291(A/T) and ERAP1rs26618(T/C) genotyping was performed using RT-PCR. Results: The frequency of the ACErs4291AA genotype was higher among the severe COVID-19 group than others (*p* < 0.001). The ERAP1rs26618TT genotype frequency was higher among the severe COVID-19 group in comparison with the mild group (*p* < 0.001) and non-infected controls (*p* = 0.0006). The frequency of the ACErs4291A allele was higher among severe COVID-19 than mild and non-infected groups (64.4% vs. 37.5%, and 34.4%, respectively), and the ERAP1rs26618T allele was also higher in the severe group (67.5% vs. 39.4%, and 49.4%). There was a statistically significant association between severe COVID-19 and ACErs4291A or ERAP1rs26618T alleles. The coexistence of ACErs4291A and ERAP1rs26618T alleles in the same individual increase the severity of the COVID-19 risk by seven times [OR (95%CI) (LL–UL) = 7.058 (3.752–13.277), *p* < 0.001). A logistic regression analysis revealed that age, male gender, non-vaccination, ACErs4291A, and ERAP1rs26618T alleles are independent risk factors for severe COVID-19. Conclusions: Persons carrying ACErs4291A and/or ERAP1rs26618T alleles are at higher risk of developing severe COVID-19.

## 1. Introduction

Coronavirus disease 2019 (COVID-19) is an ongoing pandemic caused by the newly emerging severe acute respiratory syndrome coronavirus 2 (SARS-CoV-2) that has spread across the world and has significant morbidity and mortality. Clinically, there is a great variability, ranging from asymptomatic infection to severe respiratory distress [1].

There are many factors that could affect the susceptibility and outcome of COVID-19, including viral and host factors [2]. Old age, male gender, and comorbidities were reported as independent risk factors for severe COVID-19 [1]. A change in the viral genome results in variations in cell physiology and responses to the infection [3]. Moreover, SARS-CoV-2 contagion is affected by the cellular factors [2].

Unpredictable outcomes of COVID-19, even among young adults, have raised the question of whether host genetic factors could have a role in these variable cellular responses [2].

Recent studies have proved that genetic polymorphisms of the genes affecting the pathway of SARS-CoV-2 infection and the elucidated immune response can affect the clinical manifestations and severity of COVID-19 [1,2,4,5].

COVID-19 is initiated by SARS-CoV-2 binding to an angiotensin-converting enzyme (ACE2) expressed on pneumocytes [2]. ACE belongs to the renin–angiotensin–aldosterone system (RAAS), which has a significant role in COVID-19; this enzymatic chain not only controls the pulmonary system, but also regulates the renal and circulatory systems. It has a major role in maintaining body homeostasis [4]. ACE1 regulates the expression and activity of ACE2, which allows SARS-CoV-2 to enter the pulmonary cells. Polymorphisms in ACE1 and ACE2 could affect the clinical manifestations and severity of COVID-19 [2,4]. Furthermore, Soluble ACE2 could serve as a prognostic biomarker for COVID-19 progression [5].

Moreover, changes in the host genetics that affect viral recognition and immune response could have a role in COVID-19 outcomes. Viral-infected cells present foreign proteins on HLA-I that are recognized and eliminated by CD8+T cells. The viral antigen-processing pathway in the endoplasmic reticulum (ER) depends mainly on the endoplasmic reticulum aminopeptidase 1 (ERAP1) enzyme as it trims the precursor peptides to produce the antigenic peptides that become loaded on HLA-I. Consequently, any change in ERAP1 will affect the immune response to viral peptides [6].

We hypothesized the roles of ACE and ERAP polymorphisms in the prediction of COVID-19 outcomes.

Thus, the current research aims to assess the effect of ACErs4291(A/T) and ERAP1rs26618(T/C) polymorphisms in the human genome on COVID-19 outcomes. They could be reliable prognostic markers to predict COVID-19 outcomes among different sectors of the population, particularly in the high risk groups; these include those in old age, those with chronic diseases and pregnant females.

## 2. Subjects and Methods

### 2.1. Study Design

This case-control study enrolled 240 individuals from Kafrelskeikh University Hospital, Egypt. They were divided into three groups: group (1) included 80 patients with severe COVID-19 manifestations; group (2) involved 80 patients with mild COVID-19 manifestations; and group (3) included 80 age- and sex-matched non-infected healthy individuals. The level of consciousness was determined by the Glasgow Coma Scale. Exclusion criteria were autoimmune diseases, hepatic, cardiac, and/or renal decompensation. The recorded vital signs, heart rate, and oxygen saturation were taken from files of the first admission. Inclusion criteria for COVID-19 patients were positive PCR for SARS-CoV-2. The severity of COVID-19 manifestations was judged as follows: (1) mild cases with no pneumonia on lung CT, mild clinical symptoms, and O2 saturation > 93%; (2) severe cases with respiratory rate > 30/min, severe clinical symptoms, and O2 saturation ≤ 93% at rest [7]. Non-infected persons showed negative SARS-CoV-2 RT-PCR tests with nasopharyngeal or oropharyngeal swabs and had not been infected previously. All participants were asked about receiving the COVID-19 vaccine and doses received.

### 2.2. Ethics Approval

Ethical approval for this study was granted by the Ethics Review Committee of the Faculty of Medicine at Kafrelskeikh University, Egypt (MKSU 11-12-20). The study followed the ethical guidelines of the 1975 Declaration of Helsinki without any risk to the participants. Informed consent was gathered from all participants.

### 2.3. Samples

Three ml of venous blood was drawn from all the participants, in tubes containing Ethylene diamine tetra-acetic acid (EDTA), and then stored in −80 °C to be used for Deoxyribonucleic Acid (DNA) extraction.

### 2.4. Genotyping of ACErs4291 and ERAP1rs26618 Polymorphisms

Blood samples in EDTA were used for extraction of genomic DNA using the QIAamp DNA Blood Mini Kit (Applied Biosystems, Thermo Fisher Scientific, CA, USA). Amplification of ACErs4291 (A/T) and ERAP1rs26618 (T/C) genotypes, and allelic discrimination, were carried out using TaqMan dual-labeled probes in the StepOne™ Real-Time PCR System (Applied Biosystems, Life Technologies, CA, USA), following the manufacturer’s instructions. The probes were VIC^®^-labeled (to detect ACErs4291 “A2 and ERAP1rs26618 “T” alleles), and FAM™-labeled (to detect ACErs4291 “T” and ERAP1rs26618 “C” alleles). The PCR thermal profile was 95 °C for 10 min, then 40 cycles of 92 °C for 15 s, 60 °C for 1 min, and 72 °C for 30 s, then a final extension at 72 °C for 7 min. In each run, samples without DNA were used as negative controls. Probes hybridized to the complementary sequence in the tested samples are cleaved from the dye and thus the fluorescence signal increases [8].

### 2.5. Statistical Analysis of the Data

Data were analyzed using the IBM SPSS software package version 20.0. (Armonk, NY, USA: IBM Corp). Categorical data were represented as numbers and percentages. Quantitative data were presented with mean ± SD. Student’s *t*-test, the Chi-squared test, post hoc test (Tukey), F-test, and one-way ANOVA were used for comparisons between groups. Odds ratios (OR) with 95% confidence intervals (C.I.) were used to calculate the ratio of the odds of an event in the studied group. The equilibrium of the studied samples was assessed using the Hardy–Weinberg equation. Logistic regression analysis was performed to identify the independent factor related to severe COVID-19 manifestations. The statistical significance was set at 5%.

## 3. Results

### 3.1. Subjects’ Demographic and Clinical Data

Demographic and clinical data of participants are illustrated in Table 1. There were statically significant differences in all parameters among patients with severe COVID-19 manifestations when compared with the other groups.

### 3.2. Genotypic and Allelic Determination

The Hardy–Weinberg equation (HWE) showed insignificant differences between the observed and expected genotype frequencies among the studied groups (Table 2).

Genotyping of the ACErs4291(A/T) single nucleotide polymorphism (SNP) showed a statistically significant increase in the frequency of the ACErs4291AA genotype among the severe COVID-19 group, compared to the other groups (*p* < 0.001). At the allele level, allelic discrimination revealed a marked increase in the ACErs4291A allele frequency in the severe group, which was statistically significant in comparison with the mild and non-infected groups (64.4% vs. 37.5%, and 34.4%, respectively) (*p* < 0.001) (Table 3). With respect to the ERAP1rs26618(T/C) SNP, the three genotypes (TT, CC, and TC) were expressed in all groups, however the frequency of the ERAP1rs26618TT genotype was markedly increased in the severe COVID-19 group, in comparison with the mild COVID-19 group (*p* < 0.001) and non-infected controls (*p* = 0.0006). These results were confirmed at the allele level where the T allele was significantly higher in the severe group than the mild and non-infected groups (67.5% vs. 39.4%, and 49.4%, respectively) (Table 3).

Statistical analysis revealed a significant association between the severity of COVID-19 manifestations of ACErs4291 A and/or ERAP1rs26618 T alleles (Table 4). Haplotype frequency clarified that the coexistence of ACErs4291A and ERAP1rs26618T alleles in the same individual increases the risk of severe COVID-19 by 7 times, in comparison with the other alleles [OR (95%CI) (LL–UL) = 7.058 (3.752–13.277), *p* < 0.001) (Table 5). Univariate and multivariate logistic regression analyses for the studied parameters revealed that age, male gender, non-vaccination, and ACErs4291A and ERAP1rs26618T alleles are independent risk factors for the development of severe COVID-19 (Table 6).

## 4. Discussion

The COVID-19 pandemic has resulted in a worldwide crisis, leading to overwhelming morbidities and mortalities. The outcome of those crises exhibited multiple unexpected conditions, ranging from asymptotic infections to death, without any previous alarm or known aggravating factors; some old-aged persons survived and some youths died [4].

It was hypothesized that the renin–angiotensin–aldosterone system (RAAS) has an essential role in SARS-CoV-2 infection, and its genes polymorphisms may have a role in elucidating the multi-organ and multi-system affection of COVID-19. ACE2 is a prerequisite receptor for SARS-CoV-2 infection and the ACE1 gene, a homolog of ACE2, regulates ACE2 expression and function through the regulation of angiotensin (Ang) II levels. In addition, ACE catalyzes the conversion of angiotensin I to angiotensin II, which controls blood pressure and the fluid–electrolyte balance. RAAS is involved in regulating sodium intake and potassium excretion, and immune function. Imbalance or variations in RAAS is associated with many diseases, such as hypertension, atherosclerosis, cardiovascular diseases, psoriasis, degenerative diseases, such as Alzheimer’s, diabetes mellitus, renal diseases, and end-organ damage [4].

It was stated that the spike (S) proteins of coronaviruses mediate their infection through the efficient binding of the S1 domain of the SARS-CoV S protein with ACE. They found that a soluble form of ACE could block binding between the S1 viral protein and cells. SARS-CoV replicated efficiently on ACE2-transfected cells. Thus, ACE modification can hinder the viral entry and genetic variants of ACE and could express varying affinity to the viral proteins [9].

In addition, ERAP1 is responsible for the proteolytic processing of viral peptides to be presented by MHC-I molecules to the cytotoxic T-lymphocytes (CTLs). It is responsible for trimming peptides to reach the optimum length to be uploaded on MHC class I. In some cases, ERAP1 generates MHC-I epitopes by trimming N-extended precursors to mature epitope, while in others, it destroys them by trimming epitopes that are ≥9 residues to a smaller size to bind to MHC-I. Thus, ERAP1 has an essential role in the generation and destruction of viral epitopes [3,10]. Some researchers have generated an ERAP1-deficient mice model to evaluate the *in-vivo* role of ERAP1 in the immune response to the lymphocytic choriomeningitis virus (LCMV). They noticed that the cell expression of MHC-Ia and Ib molecules decreased markedly in these mutant animals and that the mice generated defective CTLs responses against antigens presented on MHC-I. They concluded that the absence of ERAP1 reprogrammed the immune dominance hierarchy of the immune response to LCMV [3,10].

Thus, it is imperative to determine whether ACErs4291(A/T) and ERAP1rs26618(T/C) SNPs affect the outcome of COVID-19. The current study aimed to evaluate the impact of these polymorphisms in the host genome on COVID-19 outcome. The study involved COVID-19 patients with severe manifestations, others with mild symptoms, and non-infected controls. Participants’ demographic and clinical data revealed that there are significant associations between the severity of COVID-19 manifestations and old age, male gender (*p* = 0.002), and non-vaccination (*p* < 0.001). This was in partial agreement with Cai et al., (2020) [4] who have reported that old-age people (>60 years) and former smokers are more susceptible to COVID-19.

Genotyping of the ACErs4291(A/T) SNP showed a statistically significant increase in ACErs4291AA frequency among the severe COVID-19 group (*p* < 0.001). The ACErs4291A allele was present in 64.4% of patients with severe COVID-19, versus 37.5% and 34.4% of patients with mild COVID-19 and non-infected groups, respectively (*p* < 0.001). This indicates that ACErs4291A could be a bad prognostic marker for COVID-19. This can be explained by the fact that ACErs4291 (A/T) SNP causes transversion substitution in the ACE gene, which may affect its normal function and/or distribution on cells. Therefore, it was reported that genetic variants primarily affect gene expression by distracting the transcription factors’ binding. In addition, transversion is more likely to alter the amino acid sequence of proteins than transition, as it causes larger changes in the shape of the DNA backbone, disrupts binding of transcription factors, and, in turn, may have regulatory effects upon gene expression [11].

Moreover, the ERAP1 rs26618 TT genotype was markedly increased in the severe COVID-19 group (50%) versus 12.5% in the mild COVID-19 group (*p* < 0.001) and 26.3% in the non-infected controls (*p* = 0.0006). These results were confirmed at the allele level where the T allele was significantly higher in the severe group than the mild and non-infected groups (67.5% vs. 39.4%, and 49.4%, respectively). There was a statistically significant association between the severity of COVID-19 manifestations and the ACErs4291 A and/or ERAP1rs26618 T alleles. This indicates that persons carrying these alleles are at a higher risk of developing severe manifestations than those carrying the other alleles; in turn, there is a worse prognosis. A possible explanation for this is that genetic variants of ERAP1 disturb the antigen-processing pathway, resulting in alterations in the immune dominance during viral infections. This is because ERAP1 trims viral peptides to generate antigenic epitopes [3].

The statistical analysis in the current study revealed that the coexistence of ACErs4291A and ERAP1rs26618T alleles in the same individual increased the severity of COVID-19 risk by seven times [OR (95%CI) (LL–UL) = 7.058 (3.75–13.27), *p* < 0.001). The univariate and multivariate logistic regression analysis showed that age, male gender, non-vaccination, ACErs4291A, and ERAP1 rs26618T alleles are independent risk factors for severe COVID-19.

In accordance with our results, some genome-wide association studies (GWASs) have been carried out to identify the effect of ACE and ERAP polymorphisms on the disease outcomes among different populations [4,5,12,13,14]. Yamamoto et al. (2021) [4] reported that the ethnic difference in ACE1 insertion (I)/deletion (D) polymorphism can explain the apparent differences in mortality between East and West Asia. They evaluated the evidence linking COVID-19 and ACE1 polymorphisms, and found contradictory results; this is because some researchers have stated that the DD genotype adversely affects COVID-19 symptoms, while others suggest ACE1 II genotype as the risk factor. They suggested that this contradiction could be related to interactions with other genes or unexplained biochemical reactions [4].

Dou et al. (2017) [12] investigated the relationship between ACE polymorphism and the prognosis of septic shock. They observed that the frequency of ACE rs4291 TT and ACE rs4646994 DD genotypes were higher in these cases than the non-infected group. The T allele was associated with a significant elevation in the ACE activity. The fatality rate among patients with TT or DD genotypes was very high. They concluded that ACE rs4291TT and ACE rs4646994 DD genotypes may be poor prognostic factors for septic shock patients.

In addition, it was reported that ACE1 polymorphism is functionally linked to ACE2, the natural receptor for SARS-CoV virus, and polymorphisms of ACE1 and/or ACE2 are associated with hypertension, a well-known risk factor in COVID-19 [13,14]. Both have great effects on aggravating manifestations and complications of COVID-19 [15].

Alifano et al. (2020) [16] have strongly emphasized that the role of ACE2 should be kept at the heart of scientific debate on COVID-19. They suggest that there are additional roles for ethnicity, polymorphisms, behaviors, associated illness, medications, and the environment in COVID-19. Cure and Cumhur Cure (2020) [17] have stated that the binding of SARS-CoV-2 virus and ACE2 alters the RAAS, causing central stimulation, and increases sympathetic activity. Both, in turn, increase pulmonary capillary leakage, causing respiratory distress, and can trigger cardiac arrhythmias.

On the other hand, Hatami (2020) [18] conducted a meta-analysis of the ACE (I/D) genotype prevalence in COVID-19 epidemic countries. It was found that the recovery rate from COVID-19 significantly increased with high I/D allele frequency (CI 95%: 0.05–0.91, *p* = 0.027). However, no significant difference was found in the case of death rate (CI 95%: 4.5–1.04, *p* = 0.22).

Furthermore, transmembrane serine protease (TMPRSS2) is an enzyme responsible for the priming of SARS-CoV-2 (S) protein and can have a role in viral entry to the cells. Abdelsattar et al. (2022) [2] studied the role of ACE2 and TMPRSS2 SNPs in COVID-19 severity. They noticed that ACE2 rs2285666 and TMPRSS2 rs12329760 SNPs can be predictors of COVID-19 severity.

With regard to ERAP polymorphisms, Yao et al. (2016) [19] compared the frequencies of many ERAP1 SNPs among non-small cell lung carcinoma (NSCLC) patients in Chinese Han and Polish Caucasians. A significant relation was reported between all studied SNPs and NSCLC, among Chinese but not Poles (except for rs26618). They attributed this to the genetic difference. Another possible explanation may be the discrepancies in the HLA allele frequencies between Caucasians and Orientals [20]. Likewise, some antigenic peptides, presented by MHC-I, are dependent on ERAP1 trimming, while others are independent on ERAP1 and can fit MHC molecules without trimming [21].

D’Amico et al. (2021) [22] stated that when SARS-CoV-2 enters cells via the ACE2 receptor, it causes a reduction in ACE2 levels, an imbalance in RAAS, and systemic vascular resistance and blood pressure. In addition, ERAP1 has a role in RAAS regulation and MHC-I antigen processing. ERAP1 polymorphism may exacerbate SARS-CoV-2 infection, and aggravates the clinical outcome.

Stamatakis et al. (2020) [23] analyzed the proteolytic processing of S1 spike glycoprotein of SARS-CoV-2, by ERAP1, ERAP2, and IRAP. They noticed that ERAP1 was the most efficient in generating peptides of 8–11 amino acids (the optimal length for HLA-I binding). Thus, any variations in ERAP-1 will affect the pathogenesis of SARS-CoV-2 infection. Saulle et al. (2021) [24] also agreed with this, stating that ERAP1 and ERAP2 are key factors in the generation of SARS-CoV-2 antigenic peptides, presented by HLA-I.

All these data elucidate the role of ACErs4291 or ERAP1rs26618 polymorphisms in the pathogenesis of severe COVID-19 and, they should be considered prognostic markers. However, the main limitations of the current study are the small sample size, the few studied polymorphisms, the lack of comparison with the virus strain, and the effects of drug treatments on COVID-19 outcomes. Further research on a larger scale, including more comparative parameters, is recommended.

## 5. Conclusions

There is a statistically significant association between the severity of COVID-19 manifestations and any of the ACErs4291 A or ERAP1rs26618 T alleles. The coexistence of both alleles in the same individual increases the risk of COVID-19 severity by seven times. Thus, they could be screened as prognostic markers for high risk groups.

## Figures and Tables

**Table 1 diagnostics-13-00305-t001:** Comparison between the studied groups according to demographic and clinical data.

	Non-Infected (N. = 80)	Mild COVID-19 (N. = 80)	Severe COVID-19 (N. = 80)	Test of Sig.	*p*
Age (years)	45.8 ± 10	44.1 ± 12.1	47.9 ± 8	F = 2.937	0.055
Gender					
Male	27 (33.8%)	27 (33.8%)	46 (57.5%)	χ^2^ = 12.377 *	0.002 *
Female	53 (66.3%)	53 (66.3%)	34 (42.5%)		
Vaccination					
Not vaccinated	0 (0.0%)	7 (8.8%)	39 (48.8%)	χ^2^ = 69.762 *	<0.001 *
Vaccinated	80 (100%)	73 (91.3%)	41 (51.3%)		
Doses of vaccination					
1 dose	14 (17.5%)	23 (28.8%)	33 (41.3%)	χ^2^ = 107.55 *	<0.001 *
2 doses	66 (82.5%)	50 (62.5%)	8 (10%)		
Duration of symptoms					
≤14 days	–	80 (100%)	0 (0%)	χ^2^ = 160.0 *	<0.001 *
>14 days	–	0 (0%)	80 (100%)		
Level of consciousness					
GCS ≥ 12	80 (100%)	80 (100%)	54 (67.5%)	χ^2^ = 58.318 *	<0.001 *
GCS < 12	0 (0%)	0 (0%)	26 (32.5%)		
Temperature (C°)	37.1 ^b^ ± 0.4	37.2 ^b^ ± 1	38.7 ^a^ ± 0.6	F = 126.52 *	<0.001 *
Heart rate (beats/min.)	80.6 ^c^ ± 6.9	93.6 ^b^ ± 12.4	111.6 ^a^ ± 7.5	F = 227.06 *	<0.001 *
Oxygen saturation (%)	98.4 ^a^ ± 0.8	98.3 ^a^ ± 0.7	70.9 ^b^ ± 6	F = 1596.67 *	<0.001 *
Ventilator	0 (0%)	0 (0%)	26 (32.5%)	χ^2^ = 58.318 *	<0.001 *
O_2_ supply	0 (0%)	0 (0%)	54 (67.5%)	χ^2^ = 139.355	<0.001 *
Room Air	80 (100%)	80 (100%)	0 (0%)	χ^2^ = 240.0 *	<0.001 *
Past medical history	0 (0%)	26 (32.5%)	65 (81.3%)	χ^2^ = 113.672	<0.001 *

GCS: Glasgow Coma Scale, SD: Standard deviation, χ^2^: Chi square test, F: F for One way ANOVA test, pairwise comparison bet. each two groups were performed using the post hoc test (Tukey), *p*: *p* value for comparing between the three studied groups, *: Statistically significant at *p* ≤ 0.05, means with common letters are not significant (i.e., Means with Different letters are significant), past medical history included chronic conditions, such as hypertension, diabetes mellitus, and thyroid dysfunction.

**Table 2 diagnostics-13-00305-t002:** Hardy–Weinberg for different SNPs in each group.

SNPs Genotypes	Non-Infected (n = 80)	Mild COVID-19 (n = 80)	Severe COVID-19 (n = 80)
ACErs4291			
TT^®^	38 (47.5%)	35 (43.8%)	11 (13.8%)
AT	29 (36.3%)	30 (37.5%)	35 (43.8%)
AA	13 (16.3%)	15 (18.8%)	34 (42.5%)
^HW^p	*p* = 0.079	*p* = 0.074	*p* = 0.680
ERAP1 rs26618			
CC^®^	22 (27.5%)	27 (33.8%)	12 (15%)
TC	37 (46.3%)	43 (53.8%)	28 (35%)
TT	21 (26.3%)	10 (12.5%)	40 (50%)
^HW^p	*p* = 0.503	*p* = 0.260	*p* = 0.070

HW: Hardy–Weinberg equilibrium^®^: Reference group, *p*: *p* value for Chi square for goodness of fit, and level of significance at *p* ≤ 0.05.

**Table 3 diagnostics-13-00305-t003:** Comparison between the three studied groups according to different SNPs.

SNPs Genotypes and Alleles	Non-Infected (N. = 80)	Mild COVID-19 (N. = 80)	Severe COVID-19 (N. = 80)	Significance Bet. Groups
ACErs4291				
TT^®^	38 (47.5%)	35 (43.8%)	11 (13.8%)	*p*_1_ = 0.283, *p*_2_ < 0.001 *, *p*_3_ < 0.001 *
AT	29 (36.3%)	30 (37.5%)	35 (43.8%)	
AA	13 (16.3%)	15 (18.8%)	34 (42.5%)	
Allele				*p*_1_ = 0.560, *p*_2_ < 0.001 *, *p*_3_ < 0.001 *
T^®^	105 (65.6%)	100 (62.5%)	57 (35.6%)	
A	55 (34.4%)	60 (37.5%)	103 (64.4%)	
ERAP1 rs26618				
CC^®^	22 (27.5%)	27 (33.8%)	12 (15%)	*p*_1_ = 0.088, *p*_2_ = 0.006 *, *p*_3_ < 0.001 *
TC	37 (46.3%)	43 (53.8%)	28 (35%)	
TT	21 (26.3%)	10 (12.5%)	40 (50%)	
Allele				*p*_1_ = 0.072, *p*_2_ = 0.001 *, *p*_3_ < 0.001 *
C^®^	81 (50.6%)	97 (60.6%)	52 (32.5%)	
T	79 (49.4%)	63 (39.4%)	108 (67.5%)	

*p*_1_: *p* value for Chi square test for comparing between non-infected and mild groups. *p*_2_: *p* value for Chi square test for comparing between non-infected and severe groups. *p*_3_: *p* value for Chi square test for comparing between mild and severe groups. *: Statistically significant at *p* ≤ 0.05.

**Table 4 diagnostics-13-00305-t004:** Univariate analysis for severe cases regarding different SNPs, genotype and allele.

SNPs Genotypes and Alleles	Severe vs. Mild^®^		Severe vs. Non-Infected^®^	
	*p*	OR (LL–UL 95%CI)	*p*	OR (LL–UL 95%CI)
ACErs4291				
TT^®^		1.000		1.000
AT	0.002 *	3.712 (1.611–8.553)	0.001 *	4.169 (1.814–9.582)
AA	<0.001 *	7.212 (2.903–17.917)	<0.001 *	9.035 (3.577–22.824)
Allele				
T^®^		1.000		1.000
A	<0.001 *	3.012 (1.911–4.747)	<0.001 *	3.450 (2.179–5.462)
ERAP1 rs26618				
CC^®^		1.000		1.000
TC	0.367	1.465 (0.639–3.360)	0.454	1.387 (0.588–3.271)
TT	<0.001 *	9.000 (3.409–23.762)	0.005 *	3.492 (1.449–8.416)
Allele				
C^®^		1.000		1.000
T	<0.001 *	3.198 (2.022–5.057)	0.001 *	2.130 (1.353–3.351)

OR: Odd’s ratio, ^®^: Reference group, CI: Confidence interval, LL: Lower limit, UL: Upper Limit, *p*: *p* value for Univariate regression analysis for comparison with the reference genotype, *: Statistically significant at *p* ≤ 0.05.

**Table 5 diagnostics-13-00305-t005:** Haplotype frequency for ACE2 and ERAP1 in the three study groups.

ACE2 and ERAP1	Haplotype Frequencies (%)			Severe vs. Mild^®^		Severe vs. Non-Infected^®^	
	Non-Infected (n = 160)	Mild (n = 160)	Severe (n = 160)	*p* _1_	OR (95%CI) (LL–UL)	*p* _2_	OR (95%CI) (LL–UL)
TC^®^	64 (40%)	68 (42.5%)	23 (14.4%)		1.000		1.000
TT	41 (25.6%)	32 (20%)	34 (21.3%)	0.001 *	3.141 (1.598–6.174)	0.013 *	2.308 (1.194–4.458)
AC	17 (10.6%)	29 (18.1%)	29 (18.1%)	0.002 *	2.957 (1.470–5.947)	<0.001 *	4.747 (2.208–10.203)
AT	38 (23.8%)	31 (19.4%)	74 (46.3%)	<0.001 *	7.058 (3.752–13.277)	<0.001 *	5.419 (2.925–10.038)

^®^: reference group, OR: Odds ratio, CI: Confidence interval, LL: Lower limit, UL: Upper Limit, *p*_1_: *p* value for comparing between severe and mild COVID-19 patients, *p*_2_: *p* value for comparing between severe COVID-19 patients and non-infected groups, *: Statistically significant at *p* ≤ 0.05.

**Table 6 diagnostics-13-00305-t006:** Univariate and multivariate logistic regression analysis for the studied parameters affecting severe from mild (n = 80 vs. 80).

Parameter	Univariate		^#^ Multivariate	
	*p*	OR (LL–UL 95%CI)	*p*	OR (LL–UL 95%CI)
Age (every 10 years)	0.020 *	1.446 (1.061–1.971)	0.015 *	1.555 (1.091–2.216)
Male	0.003 *	2.656 (1.399–5.043)	0.002 *	3.541 (1.612–7.778)
Not vaccinated	<0.001 *	9.920 (4.070–24.176)	<0.001 *	8.57 (0.076–21.102)
ACE2rs4291 AT + AA genotypes	<0.001 *	4.879 (2.249–10.586)	0.005 *	3.576 (1.483–8.623)
ERAP1 rs26618 TC + TT genotypes	0.007 *	2.887 (1.338–6.229)	0.006 *	3.623 (1.444–9.089)

OR: Odd’s ratio, CI: Confidence interval, LL: Lower limit, UL: Upper Limit, ^#^: All variables with *p* < 0.05 was included in the multivariate, *: Statistically significant at *p* ≤ 0.05.

## Data Availability

All data are available within the article.

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
