# Peer review of "Impact of ACE and Endoplasmic Reticulum Aminopeptidases Polymorphisms on COVID-19 Outcome"

_diagnostics, 2023, doi:10.3390/diagnostics13020305_

Round 1
Reviewer 1 Report
Manuscript entitled “Impact of ACE and Endoplasmic Reticulum Aminopeptidases 2 Polymorphisms on COVID-19 Outcome”, is an excellent work in which authors have selected an unexplored and uttermost important area to identify genotypic polymorphism in ACE and Endoplasmic Reticulum Aminopeptidases correlating it with COVID19 disease. Although it is an excellent approach to correlate the genotypic polymorphisms with the viral disease, still I’m having certain suggestions to improve the quality of manuscript with intent to increase its readership.
1. The Introduction section of the manuscript is too short lacking important information related to the background of the executed research. This section needs to be revised by providing some existing information related to the polymorphic changes impacting the diseased conditions.
2. Authors have cited very few references in the manuscript and most of them are from year 2020, which needs to be updated as the data related to the COVID19 has been drastically increased revealing the better understanding of the disease.
3. Some recent references related to the COVID19 needs to be cited in the manuscript:
§ Kciuk, M., Mujwar, S., Rani, I., Munjal, K., GieleciÅ„ska, A., Kontek, R. and Shah, K., 2022. Computational Bioprospecting Guggulsterone against ADP Ribose Phosphatase of SARS-CoV-2. Molecules, 27(23), p.8287.
Author Response
We are very much thankful to you for your deep and thorough review. We earnestly appreciate your great work and constructive comments concerning our manuscript entitled " Impact of ACE and Endoplasmic Reticulum Aminopeptidases Polymorphisms on COVID-19 Outcome". All these comments are valuable and helpful for revising and improving our paper, as well as significant to our further research.
Comment no. 1: The Introduction section of the manuscript is too short lacking important information related to the background of the executed research. This section needs to be revised by providing some existing information related to the polymorphic changes impacting the diseased conditions.
- Response: The introduction has been rephrased by providing some existing information related to the polymorphic changes impacting the diseased conditions. All changes are marked in green.
Comment no. 2: Authors have cited very few references in the manuscript and most of them are from year 2020, which needs to be updated as the data related to the COVID19 has been drastically increased revealing the better understanding of the disease.
- Response: Some references have been replaced by new and more relevant ones in the revised manuscript and marked in green.
Comment no. 3:Some recent references related to the COVID19 needs to be cited in the manuscript: Kciuk, M., Mujwar, S., Rani, I., Munjal, K., Gielecińska, A., Kontek, R. and Shah, K., 2022. Computational Bioprospecting Guggulsterone against ADP Ribose Phosphatase of SARS-CoV-2. Molecules, 27(23), p.8287.
- Response: This ref. has been cited in the revised manuscript and marked in green.

Reviewer 2 Report
1. This is a single-center study and lacks novelty and does not fall under significant findings.
2. Apart from the ERAP-1 enzyme, many other enzymes significantly affect the immune response, so only ERAP-1 ACErs4291(A/T) and ERAP1rs26618(T/C) were selected.
3. In the study design, why authors did not include the samples from non-symptomatic individuals, but PCR and ELISA positive?
4. The samples were collected from a single center in Egypt, but it is not mentioned who collected the samples, how those samples were brought to Saudi Arabia, and what are the main contributions of other authors from Saudi Arabia. A general statement has been given that all authors were involved in everything, how it is this possible? I am not convinced by this statement.
5. What was the main objective of this study, it is not clearly defined.
6. This study is based on SARS-CoV-2, RNA virus, it should be converted into DNA by Reverse transcriptase enzyme (RT-PCR), but the authors have isolated genomic DNA.
7. For statistical analysis, did the authors use single or double ANOVA?
8. Table 1 is missing information about non-symptomatic individuals, they should include the samples from PCR and ELISA Positive, non-symptomatic individuals. The data should be statistically analyzed.
9. Authors should include the effect of other enzymes on immunogenic response.
10. Authors should mention the limitations of the study.
Author Response
We are very much thankful to you for your deep and thorough review. We earnestly appreciate your great work and constructive comments concerning our manuscript entitled " Impact of ACE and Endoplasmic Reticulum Aminopeptidases Polymorphisms on COVID-19 Outcome". All these comments are valuable and helpful for revising and improving our paper, as well as significant to our further research.
Comment no. 1: This is a single-center study and lacks novelty and does not fall under significant findings.
- Response: The current study aimed to evaluate the impact of ACErs4291(A/T) and ERAP1rs26618 (T/C) SNPs in the host genome on COVID-19 outcome. Most recent researches focus on the viral factors however our research focuses on the host factors. This could provide reliable prognostic markers to predict COVID-19 outcomes among different sectors of the population; particularly high-risk groups as old age, those with chronic diseases or pregnant females.
Comment no. 2: Apart from the ERAP-1 enzyme, many other enzymes significantly affect the immune response, so only ERAP-1 ACErs4291(A/T) and ERAP1rs26618(T/C) were selected.
- Response: Because ACE is involved in the viral entry to the cell and ERAP is involved in the processing of viral antigens to be presented to the immune cells. We aim to study other genes in our future research.
Comment no. 3: In the study design, why authors did not include the samples from non-symptomatic individuals, but PCR and ELISA positive?
- Response: The study design has been corrected in the whole manuscript (to be severe, mild, and non-infected persons) and marked in green. However, we cannot collect samples from persons who are positive but not symptomatic as COVID-19 ELISA or PCR tests are not mandatory screening tests in our country only those who have even mild symptoms go for investigation.
Comment no. 4: The samples were collected from a single center in Egypt, but it is not mentioned who collected the samples, how those samples were brought to Saudi Arabia, and what are the main contributions of other authors from Saudi Arabia. A general statement has been given that all authors were involved in everything, how it is this possible? I am not convinced by this statement.
- Response: Researchers from Egypt have collected the samples and performed the lab. Investigation. Detailed roles of the authors have been clarified and marked in green. Dr. Amany A. Ghazy works at Egyptian and Saudi universities but in funded research, only one affiliation is mentioned according to the policy of the funding University.
" Dr. Amany A Ghazy, Prof. Ibrahim A Taher, and Dr. Abdulrahman H Almaeen have planned the study. Acquisition of data, collection of samples and lab. Investigations were done by Dr. Amel Elsheredy and Dr. Amany A Ghazy. Data analysis and interpretation of results were performed by Prof. Ibrahim A Taher, and Dr. Abdulrahman H Almaeen, and Dr. Abdullah N. Alrasheedi. All authors have shared in the writing, revising, drafting, and final proofing of the manuscript".
Comment no. 5: What was the main objective of this study, it is not clearly defined.
- Response: This has been clarified in the revised manuscript and marked in green. "The current study aimed to evaluate the impact of ACErs4291(A/T) and ERAP1rs26618 (T/C) SNPs polymorphisms in the host genome on COVID-19 outcome".
Comment no. 6: This study is based on SARS-CoV-2, RNA virus, it should be converted into DNA by Reverse transcriptase enzyme (RT-PCR), but the authors have isolated genomic DNA.
- Response: The current study is based on host genetic polymorphism, not viral factors. This point was clarified in the revised manuscript and marked in green.
Comment no. 7: For statistical analysis, did the authors use single or double ANOVA?
- Response: One-way ANOVA and this has been clarified in the revised manuscript and marked in green.
Comment no. 8: Table 1 is missing information about non-symptomatic individuals, they should include the samples from PCR and ELISA Positive, non-symptomatic individuals. The data should be statistically analyzed.
- Response: This has been clarified in the inclusion criteria in the revised manuscript and marked in green.
Comment no. 9: Authors should include the effect of other enzymes on immunogenic response.
- Response: Another enzyme has been mentioned in the discussion of the revised manuscript and marked in green.
Comment no. 10: Authors should mention the limitations of the study.
- Response: Limitations have been mentioned in the revised manuscript and marked in green. "However, the main limitations in the current study is the small sample size, and lack of comparison with virus strain, and the effects of drug treatments on COVID-19 outcomes. Further research on larger scale including more comparative parameters is recommended.
Reviewer 3 Report
In this study, authors recruited three groups of volunteers, analyzed the ACE and ERAP1 polymorphisms, and found that the individuals with certain amino acid substitutions have a higher risk of severe symptoms after SARS-CoV-2 infection. This study is clearly presented and generally well-written. However, some flaws need to be clarified before publication.
Specific points:
1) The authors failed to state the hypothesis and to explain their experimental design in the INTRODUCTION part. The authors need to better explain why do they focus on ACErs4291(A/T) and ERAP1rs26618(T/C) polymorphisms?
2) More background information about these two polymorphism sites needs to be provided in the INTRODUCTION part. Obviously, this is based on some other studies. Therefore, more background information should be provided to facilitate the understanding for broad audiences.
3) Table 3 needs to be optimized. The p values are only provided for comparison of some data. For example, for ACErs4291, p1 value showed the difference level between control and mild groups, but only for TT, not for AT and AA allele. Similarly, p3 value showed the different level between mild and severe groups, but only for AA, not for TT and AT. Therefore, it seems that many comparisons are still lacking.
Or all these p values in Table 3 for ACErs4291 are for AA comparison among different groups. If so, the table needs to be updated and modified to present the data more precisely.
4) The authors only studies two sites in this study. I strongly recommend at least some other polymorphism sites, for example, ACE1 or ACE2 polymorphism sites, to be studied further as negative controls.
5) It is even better that the authors should amplify and perform Sauger sequencing for these following related genes: ACE, ACE1, ACE2, ERAP1, ERAP2 and so on. In this way, the analyzed results would be much less biased.
6) Line 243, “SARS-CoA” should be revised to “SARS-CoV”.
Author Response
We are very much thankful to you for your deep and thorough review. We earnestly appreciate your great work and constructive comments concerning our manuscript entitled " Impact of ACE and Endoplasmic Reticulum Aminopeptidases Polymorphisms on COVID-19 Outcome". All these comments are valuable and helpful for revising and improving our paper, as well as significant to our further research.
Comment no. 1: The authors failed to state the hypothesis and to explain their experimental design in the INTRODUCTION part. The authors need to better explain why do they focus on ACErs4291(A/T) and ERAP1rs26618(T/C) polymorphisms?
- Response: The hypothesis and why we focus on these polymorphisms were stated clearly in the revised manuscript and marked in green.
Comment no. 2: More background information about these two polymorphism sites needs to be provided in the INTRODUCTION part. Obviously, this is based on some other studies. Therefore, more background information should be provided to facilitate the understanding for broad audiences.
- Response: The introduction has been rephrased by providing more information related to these polymorphisms and their expected impact on the diseased conditions.
Comment no. 3: Table 3 needs to be optimized. The p values are only provided for comparison of some data. For example, for ACErs4291, p1 value showed the difference level between control and mild groups, but only for TT, not for AT and AA allele. Similarly, p3 value showed the different level between mild and severe groups, but only for AA, not for TT and AT. Therefore, it seems that many comparisons are still lacking. Or all these p values in Table 3 for ACErs4291 are for AA comparison among different groups. If so, the table needs to be updated and modified to present the data more precisely.
- Response: Table 3 illustrates the distribution of the 3 alleles for each SNP in the studied groups, p1 indicates the p value for Chi square test for comparing the distribution of the 3 alleles between non-infected and mild groups, p2 indicates the p value for chi square test for comparing the distribution of the 3 alleles between non-infected and severe groups, and p3 indicates the p value for chi square test for comparing the distribution of the 3 alleles between mild and severe groups. However, table 4 is univariate analysis for significant difference regarding each allele.
Comment no. 4: The authors only studies two sites in this study. I strongly recommend at least some other polymorphism sites, for example, ACE1 or ACE2 polymorphism sites, to be studied further as negative controls.
- Response: This is an interesting point and we will consider it in our further research but for the current research the samples were taken long time ago and we cannot perform these tests right now and the results will be inaccurate or missed. This point is mentioned in the limitation of the study in the revised manuscript and marked in green.
Comment no. 5: It is even better that the authors should amplify and perform Sauger sequencing for the following related genes: ACE, ACE1, ACE2, ERAP1, ERAP2 and so on. In this way, the analyzed results would be much less biased.
- Response: This is an interesting point and we will consider it in our further research but for the current research the samples were taken a long time ago and we cannot perform these tests right now and the results will be inaccurate or missed.
Comment no. 6: Line 243, “SARS-CoA” should be revised to “SARS-CoV”.
- Response: Corrected in the revised manuscript and marked in green.
Round 2
Reviewer 2 Report
(1) Affiliation 1: correct it, its written And nd......
(2) Page no 3: Covid 19- , PLEASE CORRECT IT.
(3) Font size of table 2 and 3, make uniform.
Reviewer 3 Report
The manuscript introduction part has been largely improved. No further comments.